# Walking Speed and Risk of Falling Patients Operated for Selected Malignant Tumors

**DOI:** 10.3390/healthcare11233069

**Published:** 2023-11-30

**Authors:** Anna Latajka, Małgorzata Stefańska, Marek Woźniewski, Iwona Malicka

**Affiliations:** 1Lower Silesian Oncology, Pulmonology, and Hematology Center, 53-413 Wroclaw, Poland; latajka.anna@dco.com.pl; 2Department of Physiotherapy, Wroclaw University of Health and Sport Sciences, 51-612 Wroclaw, Poland; malgorzata.stefanska@awf.wroc.pl (M.S.); marek.wozniewski@awf.wroc.pl (M.W.)

**Keywords:** cancer, surgical treatment, walking speed, fall risk

## Abstract

Background: A literature review reveals that studies on walking and fall occurrences in the context of cancer have predominantly centered on geriatric patients. Nonetheless, cancer patients of all ages are susceptible to such risks. Both cancer and its treatments contribute to significant risk factors for disturbances in walking and falls, encompassing muscle weakness, impaired balance, reduced proprioception, cognitive impairment, and functional limitations. Aim: to assess walking speed and the risk of falls among patients undergoing surgery for the most common malignancies: breast (BU), lung (P), colorectal (DS), and reproductive organs (G). Material and Methods: An observational study was conducted using a cohort design. A total of 176 individuals participated in the study, including 139 cancer patients, who were divided into four groups: BU (N = 30), P (N = 35), DS (N = 35), and G (N = 39), as well as 37 healthy volunteers in the control group (C, N = 37). All participants underwent an assessment of walking speed using BTS G-WALK^®^ and an evaluation of the number of falls and the risk of falling using a Fall Control Card. Results: There was a significant decrease in walking speed after surgery compared to the time before surgery, from 2.7% in the BU group, through 9.3% in the P group, and 19.2% in the DS group, to 30.0% in the G group. At the same time, for groups G and DS, the average walking speed fell below 1.0 m/s, amounting to 0.84 m/s and 0.97 m/s, respectively, in the measurement after the surgery and 0.95 m/s and 1.0 m/s in the follow-up measurement. Falling occurred in all the groups except for the BU group. The created logistic regression model showed that increasing the walking speed measured after the procedure (study 2) by 1 m/s reduces the risk of falling by approximately 500 times (OR = 0.002). Limitations in daily activity were observed in the follow-up examination (study 3) in 75% of patients. Conclusions: Surgical intervention has an impact on walking speed, and being part of the study group influences the risk of falling. Further research is needed to determine the precise risk of falls in cancer patients.

## 1. Introduction

Malignant tumors are one of the most common groups of diseases in developed countries, causing long-term effects that affect patients’ lives, changing their quality in every sphere: personal, professional, and social [1]. In women, the highest incidence of malignant tumors is recorded between the ages of 50 and 74. These include breast (22.9%), lung (9.9%), endometrial (7.0%), colon (5.9%), and ovarian (4.3%) cancer, while in men, between 55 and 79 years of age, they are prostate (20.6%), lung (16.1%), colon (6.8%), bladder (6.4%), rectum (4.2%), and stomach (3.8%) cancer [2].

A literature review reveals that studies on walking and fall occurrences in the context of cancer have predominantly centered on geriatric patients. Nonetheless, cancer patients of all ages are susceptible to such risks. Both cancer and its treatments contribute to significant risk factors for disturbances in walking and falls, encompassing muscle weakness, impaired balance, reduced proprioception, cognitive impairment, and functional limitations [3,4,5,6].

Surgery is the leading treatment method for solid tumors in approximately 80% of cancer patients [7]. Injury to the body resulting from surgical procedures can lead to the onset of pain, which, in turn, may cause anxiety. The manifestations of anxiety and pain are physiological patterns that closely resemble each other, making it challenging to interpret them as unique to either pain or anxiety. Dizziness, restlessness, and muscle tremors can be symptoms indicative of both anxiety and pain simultaneously. Prolonged pain can trigger or exacerbate depressive disorders [8,9]. Research suggests that individuals with depression exhibit deviations in their walking [10].

Surgery may cause functional complications within the musculoskeletal system, affecting a reduction in gait efficiency. In addition, as a result of surgical treatment, functional disorders of the musculoskeletal system appear, resulting primarily from aspects related to the formation of scars and adhesions within soft tissues, which cause a limitation of mobility in the joints, pain, and further restriction of physical activity. There are changes in nerve conduction caused by damage to the skin structures, subcutaneous tissue, fascia, and muscles. Body posture disorders may result from increased postoperative pain, changes in joint mechanics, and overloading of muscles and tissues. Changing the mechanics of movement in the joints makes it uneconomical and leads to faster tissue wear and the appearance of degenerative changes [11]. 

Patients with cancer often use multiple medications simultaneously, and various drug types like opiates, benzodiazepines, steroids, antipsychotics, and sedatives are linked to a substantial risk of falls. Additionally, patients with cancer undergo distinctive forms of treatment, including radiation therapy, chemotherapy, and biologic response modifiers. Those receiving radiotherapy frequently report experiencing weakness and fatigue, and this fatigue can potentially contribute to falls in cancer patients. Chemotherapy also poses a risk for falls in this patient population, with the risk increasing as the cumulative dose of chemotherapy and the use of neurotoxic drugs rise [12].

A disturbed gait pattern may limit not only the efficiency of walking and cause an increase in energy expenditure but also the appearance of secondary, incorrect compensatory reactions that may become permanent [13] and consequently contribute to a greater risk of falls [14]. Lowry et al. [15] indicate that the typical walking speed of healthy adults is in the range of 1.2 to 1.4 m/s and begins to decline naturally between the fifth and sixth decades of life. Maintaining a speed above 1.0 m/s is associated with greater independence in activities of daily living and a lower risk of hospitalization due to adverse events—including falls. Quach et al. [16] and Van Kan et al. [17] showed that the lowest risk of falling occurs at a walking speed of 1–1.3 m/s. The critical value is considered to be 0.6 m/s. Every 0.2 m/s increase in speed from the critical value reduces the risk of requiring personal care by 38% [18].

Therefore, the study aimed to assess the walking speed and risk of falling among patients undergoing surgery for the most common malignancies: breast, lung, colorectal, and reproductive organs.

## 2. Materials and Methods

### 2.1. Participants and Recruitment

An observational study was conducted using a cohort design (STROBE Statement in the Appendix A). 

The study included, from 14 November 2018 to 3 January 2022, 139 cancer patients and 37 healthy volunteers who met the inclusion criteria. Four patient cohorts—breast (BU), reproductive organ (G), pulmonary (P), and digestive system (DS)—were derived from a consecutive series of cancer patients qualified for surgical treatment at the Lower Silesian Oncology, Pulmonology, and Hematology Center in Wroclaw, Poland.

Group 1—BU—breasts (Breast Unit)—30 people were examined: 30 women and 0 men. Surgical procedures in the chest area (simple mastectomy, mastectomy with simultaneous implant reconstruction, mastectomy with lymphadenectomy).

Group 2—G—gynecological—35 people were examined: 35 women and 0 men. Surgical procedures within the pelvic area (gynecological operations—oncological operations of the reproductive organ in women with the opening of the lower abdominal cavity—gynecological laparotomy)

Group 3—P—pulmonary—39 people were examined: 18 women and 21 men. Thoracic surgery with resection of a part of the lung—operator’s access to the lung tissue from the intercostal area, VATS lobectomy.

Group 4—DS—digestive system—35 people were examined: 12 women and 23 men. Abdominal surgeries—laparotomy for colorectal cancer. 

Group 5—C—control group—37 people were examined: 29 women and 8 men. Healthy people, not suffering from malignant tumors in the past or currently.

The inclusion criteria for the study were as follows:Age: 50–70.BMI 20–35.Independent movement without the use of orthopedic aids.No fall in the last 3 months.Informed consent of the patient to participate in the study.

The criteria for exclusion were as follows:Neoadjuvant chemotherapy and radiotherapy and postoperative complications, e.g., massive hematomas, revisions of postoperative wounds, and wound infections.Orthopedic and traumatological diseases that disturb the normal gait pattern.Diseases of the nervous system.Known balance and coordination disorders.Psychiatric disorders, including a history of diagnosed and treated depression.The use of medicines that affect psychophysical efficiency.Difficult cooperation with the examined person and refusal to participate in the study (at every stage).

Patients were assigned to each group (BU, G, P, DS) using a computer program that randomly selected patients from each patient cohort while considering the inclusion and exclusion criteria as well as the type of surgery.

During the stay in the hospital, each patient was examined twice: before the surgery (study 1) and after the surgery (study 2, when they were able to stand up and move on their own—usually on the second to fourth day after the surgery). In addition, a follow-up examination was conducted four weeks after the end of hospitalization (study 3). Throughout the study, patients were required to maintain their usual lifestyle.

All patients received medical and physiotherapeutic care.

### 2.2. Research Methods

#### 2.2.1. Walking Speed Measurement

Gait assessment was performed using the BTS G-WALK^®^ accelerometer (BTS Bioengineering, Milan, Italy). A sensor (G-Walk-sensor) was placed on the patient’s body using a belt at the level of the lumbar spine (according to the manufacturer’s recommendations: in the area of the intervertebral space L4–L5), and then the patient’s task was to follow the researcher’s command: “start” using your own speed for a distance of 20 m in a straight line one way, and after 20 m, turn back to the starting point. The same footwear was worn during all the examinations of a given patient. Then, the data collected via Bluetooth were sent to a computer and processed using the BTS-WALK software dedicated to the device, and the walking speed was analyzed.

G-Walk is a wireless system consisting of an inertial sensor composed of a triaxial accelerometer, a magnetic sensor, and a triaxial gyroscope. The device calibrates itself automatically each time the walking testing function is activated for a specific patient.

Research indicates that the BTS G-WALK^®^ sensor system is reliable for all measured spatiotemporal parameters. The intraclass correlation coefficient values for walking speed were excellent between consecutive measurements on the same day, with values ranging from 0.83 to 0.96. In terms of validity, the intra-class correlation coefficient values between measurement systems showed excellent levels of agreement for walking speed; range = 0.98 to 0.99 [19].

#### 2.2.2. Fall Control Card

The Fall Control Card allows you to observe falls and subjective feelings related to your daily physical activity for 1 month. 

A fall was defined as an event in which an adult unintentionally came to rest on the ground or other lower supporting surfaces, unrelated to a medical incident or to an overwhelming external physical force.

The falls were categorized based on their causes (standing up, sitting down, walking, bending, and turning/turning around), consequences (hospitalization, need for medical assistance, sufficient assistance from others, independent recovery, and resumption of activities), and circumstances (at home/outside the home).

The Fall Control Card was distributed solely to patients in the study group in the form of a self-completed diary.

According to the answers to the questions contained in the card, the researcher’s task was to collect information about possible falls and analyze them. 

### 2.3. Ethics

The study received a positive opinion from the Senate Committee on Research Ethics at Wroclaw University of Health and Sport Sciences. Consent number: 28/2018. Approval date: 14 September 2018.

The study was approved and was registered in the Lower Silesian Center of Oncology, Pulmonology, and Hematology in Wroclaw, Poland. Consent number: NDBI-106/18, NDBI 4/18/BN. 

### 2.4. Statistical Analysis

Arithmetic means and standard deviations were calculated for measurable variables. The frequency of their occurrence (percentage) was calculated for categorical variables. The normality of the distribution was checked with the Shapiro–Wilk test, and the homogeneity of the variance was checked with the Levene test. An ANOVA for repeated measures was applied with a post hoc comparison using the Tukey test for quantitative variables. Test power was calculated. The Chi-square (χ^2^) test was used for nominal variables. A logistic regression model was performed, and the odds ratio (OR) for the risk of falls was calculated. The effect size of the ANOVA was calculated from Eta-squared (ƞ^2^) and then transformed to Cohen’s d value [20]. Values of Cohen’s d test ≥0.8 indicated great strength of the observed effect, ≥0.5 indicated a moderate effect, ≥0.2 indicated a weak effect, and <0.02 indicated no effect [21]. Cramer’s V coefficient was used to calculate the effect size of the χ^2^ test with more than one degree of freedom (categorical variables). Cramer’s V value is in the range of 0–1. The closer it is to 0, the weaker the strength of the relationship between the examined features, and the closer it is to +1, the stronger the strength of the studied relationship is [22]. Calculations were made in Statistica 13.3, PQ Stat 1.8.2, and statistical calculators at http://www.psychometrica.de/effect_size (accessed on 26 October 2023). The significance level was taken as *p* < 0.05.

## 3. Results

### 3.1. Participants Characteristics

The average age of the examined people was 60.56 ± 5.11 years, the average body height was 166.34 ± 6.74 cm, and the average body weight was 74.46 ± 12.85 kg. The BMI index for the examined people was 26.86 ± 4.08. Detailed results for individual groups are presented in Table 1. The study groups were considered homogeneous regarding age and somatic characteristics. There were no significant differences in the main effect for age and BMI (Table 1).

### 3.2. Walking speed [m/s]

Table 2 shows the mean values and standard deviation of walking speed for individual groups and subsequent measurements. Significant differences on the level of main effects were found; significant differences were found for the walking speed of the examined groups in relation to the control group and between the individual groups (Table 3); and significant differences were found for subsequent measurements in the study groups (Table 4). A significant decrease in walking speed was found for all the study groups, except BU, in the second measurement compared to the first, and a significant increase in walking speed for group G in the third measurement compared to the second measurement was found (Table 4). Significant differences between measurements 1 and 3 were shown for groups G and DS (Table 4).

### 3.3. Risk of Falling

A fall occurred in all groups except for the BU group. It was a statistically significant difference. The number of falls did not differ significantly in the study groups. A detailed analysis of the number of falls is presented in Table 5.

Most often, a fall occurred in the patient’s place of residence. Only for two patients from group G and two from group DS did the fall occur outside their home. In all the study groups, falls were most often observed while standing up, bending down, and turning. No statistically significant differences were found. 

In each group, many fell while wearing shoes with an open heel (about 70% of the falls). No statistically significant differences were found. As a consequence of a fall, the help of other people was needed in about 50% of cases. In three quarters of the people who fell, there was no injury, and one quarter reported minor injuries, e.g., epidermal abrasions. No statistically significant differences were found. Limitations in daily activity were observed in the follow-up examination (study 3) in 75% of patients. No statistically significant differences were found between the groups.

The multivariant analysis confirmed a significant impact of walking speed on the risk of falling. The analysis took into consideration belonging to a specific group, age, the BMI of the subjects, and the walking speed measured in study 2 (after surgery) or the difference between the walking speed in studies 1 and 2 (before–after surgery). The created logistic regression model showed that increasing the walking speed measured after the procedure (study 2) by 1 m/s reduces the risk of falling by approximately 500 times (OR = 0.002) (Table 6). It has also been shown that increasing the difference in walking speed measured in studies 1 and 2 by 1 m/s increases the risk of falling by approximately 200 times (OR = 200.36) (Table 7). Both regression models showed a significant effect of surgery on the risk of falls. Belonging to the study group determines an increase in risk by approximately two times.

## 4. Discussion

Recovery after surgery is understood as the time when a person strives to regain independence and, consequently, returns to everyday activities. While it is easy to determine the beginning of recovery after a surgical intervention, its end remains uncertain. Several factors influence recovery after surgery. These factors include physical symptoms, emotional disturbances, previous medical history, and its impact on recovery time at that time. Great importance is now also attached to the existence or lack of adequate and regular information and support provided to the patient by their family and/or health professionals. The extent of the surgical procedure also plays a vital role in recovery [23,24].

Gait is considered one of the most reliable parameters reflecting the general condition of a patient and a predisposing factor for safe functioning in everyday life [25]. Walking speed serves as an indicator of frailty, and its evaluation in oncology clinics can significantly enhance patient assessment, prognostication, and the customization of care [26]. In the study groups, a lower walking speed was observed in the initial measurement before surgery for the DS, G and BU groups, and this difference was also maintained in the follow-up measurement. Moreover, in all the examined groups, a significant decrease in walking speed after the surgery was observed in relation to the time before the surgery, from 2.7% in the BU group, through 9.3% in the P group, and 19.2% in the DS group, to 30.0% in the G group. At the same time, for groups G and DS, the average walking speed fell below 1.0 m/s, amounting to 0.84 m/s and 0.97 m/s, respectively, in the measurement after surgery and 0.95 m/s and 1.0 m/s in the follow-up measurement. Hence, the most substantial limitations were observed in patients who underwent surgical procedures for digestive and gynecological cancers. Surgical interventions in the abdominal and lower abdominal regions can lead to limited hip mobility and flexion contracture. Hip flexion contracture results in increased pelvic anteversion during the support phase and, in conjunction with postoperative pain, may lead to crouched walking. Furthermore, a diminished range of hip movement is correlated with a reduction in walking speed [27,28].

Walking speed predicts the length of a hospital stay, readmission, and risk of death [29]. Walking speed below 1.0 m/s is associated with a higher risk of falling [16,17], and below 0.8 m/s is an independent predictor of death in elderly cancer survivors [3]. In connection with the above, it is worth paying particular attention to group G, which seems to be at the highest risk of falling immediately after surgery and worse functioning, which may translate into survival time.

Conversely, it appears that groups P and BU face the lowest level of risk. Although the changes were statistically significant, they were of a smaller magnitude. The damage to anatomical structures during surgery may play a role in dysfunctions of the upper body quadrants. Imbalances in chest wall muscle function and the subsequent disruption of postural muscle balance could lead to increased thoracic kyphosis and subsequent lumbar lordosis, potentially contributing to lower back pain. Lower back pain is frequently accompanied by alterations in walking [28,30]. The walking speed in both groups was, however, above 1.0 m/s, which is consistent with the results obtained by other authors [31,32].

Taking into account the walking speed, Pererai et al. [33] and Middleton et al. [34] further suggest that a change in walking speed of 0.05 m/s, although small, is clinically significant, while a change in walking speed of 0.10 m/s is significant for mobility. Therefore, surgical treatment, especially in the first postoperative period, brings a substantial change in the mobility of cancer patients, indicating the need to introduce physiotherapeutic procedures adequate for existing disorders.

Bluethmann et al. further observed a recurring but varying effect of the history of cancer on patients’ mobility. A higher percentage of elderly people with a history of malignancy than those without a history of cancer used mobility aids. At the same time, the use of mobility aids varied depending on the type of cancer, with the highest rates in the group of people who suffered from breast cancer, colorectal cancer, and cancer of the reproductive organs. Cancer patients were also more likely to show signs of mobility impairment [35]. These results suggest the importance of assessing fall risk at follow-up visits and identifying risk factors that appear to be modifiable with appropriate treatment interventions. In our study, a fall occurred in all the study groups except for the BU group. Falls are a multifactorial consequence. Among patients undergoing cancer treatment, there is a deficiency in endurance, muscle weakness, and pain. Pain, as evaluated by Mata et al. [23], is the most critical risk factor for falls compared to other factors. The magnitude and extent of postoperative pain depend on the type of surgery. The most intense pain is experienced following thoracotomies and procedures in the abdominal region [36], which may explain the occurrence of falls in these three study groups (P, DS, G).

It is worth noting that most often, falls occurred in the patient’s place of residence. Li et al. emphasize that indoor falls occur more often in frail people who avoid leaving the house [37]. Falls occurred most often while standing up, bending down, or making a turn/turning. Bartoszek et al. indicated that the most common cause of falls is usually everyday activities, such as walking or changing position [38]. Based on the conducted research, it seems crucial to extend the physiotherapeutic procedure for the time after leaving the hospital. It is particularly important that the mean age of the examined patients was 60.56 ± 5.11 years. This means that gait pattern disorders and a higher risk of falling may occur in patients operated on for the most common malignancies much earlier than indicated by the literature review [39,40,41]. It is recognized that falls are a serious problem among older people; the incidence of falls increases with age [42]. In patients treated for malignant tumors, the literature review also indicates disorders resulting from combination therapy [43,44]. Our own research indicates that it can be assumed with a high probability that the examined patients were then qualified for adjuvant treatment in the form of chemotherapy, radiotherapy, or hormone therapy.

### 4.1. Strengths and Limitations of This Study

Our own analysis also indicated limitations resulting, among others, from the lack of specification of the time and type of physiotherapy applied immediately after surgery. An analysis with regard to the verticalization day was also not included, which, depending on the surgical procedure, varied from 1 to 3 days. Postoperative rehabilitation may affect the quality of gait in the first postoperative days and, consequently, the risk of falls.

The analysis also did not consider the division into sex, and the impact of gender on the risk of falling was not assessed. Initially, falls are more common in women in early old age, while in late old age, these incidents have the same frequency in both women and men. On the other hand, men are more likely to die from a fall [45]. Perhaps these differences in falls and their consequences would also be observed in the group of patients operated on for selected malignancies. 

The analysis also did not take into account the mental state of patients undergoing surgery; anxiety and depression may affect the speed of walking and, thus, the risk of falling.

Despite the indicated limitations, the results presented in the paper may contribute to improving the standards of physiotherapy for patients treated for malignant tumors, along with creating a strategy to minimize the risk of collapse for this population, which has been growing in recent years. 

### 4.2. Future Research Directions

The above knowledge of gait speed disorders, depending on the extent of the procedure and the operating site, will help develop rehabilitation methods for patients, emphasizing improving balance and coordination.

Reducing the risk of falls will significantly increase patients’ physical activity after surgical procedures due to malignant tumors, which is of particular importance in preventing the adverse effects of oncological treatment. Maintaining physical activity at an appropriate level will allow patients to adopt an active attitude in the fight against cancer and its consequences. 

In the future, it is necessary for patients operated on due to malignant tumors to strive for independence. Reducing dependence on the help of others in everyday life will enable oncological patients to use the potential of oncological treatment more optimally. It will also have a positive impact on the prognosis.

Monitoring the level of physical activity and a systematic assessment of the risk of falling during the treatment of malignant tumors and during follow-up examinations after treatment should be a standard of therapeutic management as an element necessary to achieving success and improving the quality of life and functioning in the community of patients with a history of malignant tumor treatment.

## 5. Conclusions

In conclusion, surgical procedures for cancer have an impact on the walking speed of patients being treated for malignant tumors, and the affiliation with the study group determines the risk of falling. Further research is needed to accurately determine the risk of falls among cancer patients.

## Figures and Tables

**Table 1 healthcare-11-03069-t001:** Statistical characteristics of age and somatic features.

Characteristic	G N = 35		DS N = 35		P N = 39		BU N = 30		C N = 37		ANOVA *p*	Cohen’s d
	Mean	SD	Mean	SD	Mean	SD	Mean	SD	Mean	SD		
Age (years)	60.11	4.87	61.54	5.28	60.67	5.30	61.30	4.21	59.22	5.89	0.3307	0.35
BMI (kg/m^2^)	27.47	4.57	27.03	3.41	26.32	3.95	26.05	4.38	27.44	4.11	0.4884	0.29

Groups: G—gynecological, DS—digestive system, P—pulmonary, BU—breasts, and C—control.

**Table 2 healthcare-11-03069-t002:** Walking speed and analysis of variance for repeated measurements.

	Walking Speed (m/s)						ANOVA					
	Study 1		Study 2		Study 3		Repetition			Group		
Group	Mean	SD	Mean	SD	Mean	SD	*p*	d Cohen’s	Test Power	*p*	d Cohen’s	Test Power
All	1.23	0.21	1.08	0.28	1.14	0.24						
BU	1.12	0.18	1.02	0.20	1.08	0.18	<0.0001 *	1.22	1.00	<0.0001 *	1.57	1.00
G	1.20	0.16	0.84	0.22	0.95	0.16						
DS	1.20	0.22	0.97	0.24	1.01	0.18						
P	1.29	0.23	1.17	0.23	1.26	0.19						
C	1.33	0.21	1.37	0.19	1.35	0.19						

Groups: G—gynecological, DS—digestive system, P—pulmonary, BU—breasts, and C—control; * *p* < 0.05.

**Table 3 healthcare-11-03069-t003:** Evaluation of the differences in average values of walking speed between groups, performed with Tukey’s post hoc test.

Group		Study 1				Group		Study 2				Group		Study 3			
		BU	G	DS	P			BU	G	DS	P			BU	G	DS	P
Study 1	G	0.9728				Study 2	G	0.0270 *				Study 3	G	0.4451			
	DS	0.9736	1.0000				DS	0.9992	0.3045				DS	0.9902	0.9968		
	P	0.0512	0.7943	0.7907			P	0.2130	<0.0001 *	0.0019			P	0.0398	<0.0001 *	<0.0001 *	
	C	0.0049 *	0.2989	0.2953	1.0000		C	<0.0001 *	<0.0001 *	<0.0001 *	0.0027 *		C	0.0001	<0.0001 *	<0.0001 *	0.8987

Groups: G—gynecological, DS—digestive system, P—pulmonary, BU—breasts, and C—control; * *p* < 0.05.

**Table 4 healthcare-11-03069-t004:** Evaluation of the differences in average values of walking speed between successive measurements in the study groups, performed with Tukey’s post hoc test.

Group	Study 1 vs. 2	Study 1 vs. 3	Study 2 vs. 3
G	<0.0001 *	<0.0001 *	0.0111 *
DS	<0.0001 *	<0.0001 *	0.9785
P	0.0024 *	0.9994	0.0996
BU	0.2090	0.9994	0.8875
C	0.9904	1.0000	1.0000

Groups: G—gynecological, DS—digestive system, P—pulmonary, BU—breasts, and C—control; * *p* < 0.05.

**Table 5 healthcare-11-03069-t005:** Analysis of the occurrence of a fall in the study groups (1: yes; 0: no) and the number of falls in the study groups (numerical value from 1 to 3).

	Fall				Number of Falls				
Group	No	Yes	χ^2^	Cramer’s V	1	2	3	χ^2^	Cramer’s V
BU	30	0	<0.0001 *	0.39	0	0	0	0.5427	0.31
	100%	0%			-	-	-		
G	24	11			4	6	1		
	69%	31%			36%	55%	9%		
DS	23	12			9	3	0		
	66%	34%			75%	25%	0%		
P	28	11			9	2	0		
	72%	28%			82%	18%	0%		
C	37	0			0	0	0		
	100%	0%			-	-	-		

Groups: G—gynecological, DS—digestive system, P—pulmonary, BU—breasts, and C—control; χ^2^—Chi-square test; * *p* < 0.05.

**Table 6 healthcare-11-03069-t006:** Logistic regression model indicating factors that have a significant impact on the risk of falling (walking speed).

Dependent Variable: 0—Fall No 1—Fall Yes	Coef. Β	Error Β	Wald Test	OR	95% CI	*p*
Group	6.54	3.47	3.55	2.36	1.44–3.87	0.0007 *
Age (year)	0.89	0.25	13.02	0.92	0.84–1.02	0.1157
BMI (kg/m^2^)	−0.08	0.05	2.50	1.03	0.91–1.17	0.6656
Walking speed 2 (m/s)	0.03	0.06	0.18	0.002	<0.001–0.02	<0.0001 *
*p*	<0.0001 *					
Pseudo R2	0.38					

OR—odds ratio; 95%CI—95% confidence interval; Group: 0—control; 1—breasts; 2—gynecological; 3—pulmonary; 4—digestive system; * *p* < 0.05.

**Table 7 healthcare-11-03069-t007:** Logistic regression model indicating factors that have a significant impact on the risk of falling (difference in walking speed).

Dependent Variable: 0—No Fall 1—Fall	Coef. Β	Error Β	Wald Test	OR	95%CI	*p*
Group	0.81	0.23	12.03	2.25	1.42–3.57	0.0005 *
Age (year)	−0.03	0.05	0.43	0.97	0.88–1.06	0.4233
BMI (kg/m^2^)	0.08	0.06	1.67	1.09	0.96–1.23	0.1675
Walking speed 1–2 (m/s)	5.30	1.32	16.17	200.21	15.12–2650.36	0.0001 *
*p*	<0.0001 *					
Pseudo R2	0.34					

OR—odds ratio; 95%CI—95% confidence interval; Group: 0—control; 1—breasts; 2—gynecological; 3—pulmonary; 4—digestive system; * *p* < 0.05.

## Data Availability

The data are available upon reasonable request from the corresponding author.

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
