# Peer review of "Walking Speed and Risk of Falling Patients Operated for Selected Malignant Tumors"

_healthcare, 2023, doi:10.3390/healthcare11233069_

Round 1
Reviewer 1 Report
Comments and Suggestions for Authors
The study aimed to assess the effect of surgical treatment of the most common malignant tumors on walking speed and the risk of falling. I think this study has some merits and limitations, but it requires major revisions before it can be considered for publication.
Please find specific comments below.
1. Introduction section needs a significant improvement.
The authors did not elucidate clearly why they chose to focus on four malignancies (breast, lung, colorectal and reproductive organ cancers) in their study. They should add their relative frequency, impact, or relevance to their research question. In addition, they should provide some statistics or evidence to support their claims. They could cite some sources that report the incidence, prevalence, mortality, and burden of cancer in different regions or countries. Is there no existing studies examining the relationship between surgical treatment, walking speed, and fall risk? The authors should provide more detail on previous studies that have addressed this topic, summarising the key findings and limitations of these studies and how this study addresses these issues.
2. Methods section is vague and lacks a lot of key information.
First, the authors used healthy people as a control group but not patients who did not undergo surgery as a control group. This makes it difficult to isolate the causal effect of surgery on the outcome. They should explain why they chose this design and how they ensured the comparability of the groups. Secondly,how did the authors randomise the participants for this study? Authors should provide a detailed description of the randomisation procedures and allocation hiding methods used in the manuscript. If the authors were completely randomly selected participants, what was done to ensure that they were sufficiently representative, this should also be elaborated in the manuscript. Thirdly, how did they standardise the gait assessment using the G-Walk accelerometer? The authors should provide more information on the validity and reliability of this device and software for measuring walking speed, as well as details on how they calibrated the device, instructed the participants, and controlled for any confounding factors that might affect gait performance during the study. Fourthly, how did they administer the Falls Control Card to the participants? The authors should elaborate on how they collected and verified information about possible falls over a one-month period, as well as subjective feelings related to daily physical activity. They should also explain how they defined a fall and how they classified falls according to their causes, consequences and circumstances. Finally, authors should provide the clinical registration number and ethical review approval number in their methods section. I only found an Institutional Review Board Statement at the end of the manuscript and did not see any clinical registration number. If possible, I would suggest that they report any ethical issues or challenges they encountered during their study and how they resolved them. For example, how did they ensure the safety and well-being of their participants, especially those who underwent surgery for malignant tumors? Providing these information is essential for the quality and credibility of the paper and for the readers to evaluate the ethical conduct of the study.
3. Results section has some gaps and errors that need to be corrected.
First, authors should add units to the data in tables, such as BMI in Table 1. This will make their data more clear and understandable for readers. Secondly, footnotes need to be added to tables, such as what Study 1 represents. This will help readers follow logic
and design of their study. Thirdly, * in table already represents p<0.05, so why is it written in table as p<0.0001? This is redundant and confusing. They should either use * or exact p-value, but not both. They should also use consistent format for reporting p-values throughout paper, such as using two or three decimal places. Fourthly, data in Table 3 and Table 4 are difficult to understand, with only p-values in one table. They should provide some descriptive statistics, such as means and standard deviations, for each group and each time point. They should also report effect sizes and confidence intervals for their tests of significance. This will allow readers to evaluate magnitude and precision of their results. Fifthly, what is ch2 in Table 5 and ±95CI in Table 6? They should check writing throughout. They should use proper symbols and abbreviations for their statistical terms, such as χ2 and ± . They should also define any abbreviations that they use in their paper, such as OR (odds ratio) and CI (confidence interval). Sixthly, value of OR for walking speed in Table 6 is 0.002, while value of OR for walking speed in Table 7 is 333.96. First of all, these two results are a bit suspicious. In addition, p-value is surprisingly same for both. There is a relationship between p-value and OR-value. I cannot understand why an OR value of 0.002 would show such a small p-value. They should explain how they calculated OR and p-value for walking speed and check for any errors or outliers in their data or analysis. Finally, 95% confidence range for OR value of walking speed in Table 7 is 32.96-3844.27. It is clear that maximum and minimum values are very much separated from each other. Therefore, OR value may not be credible. They should report standard error or standard deviation of OR value and check for any sources of variability or uncertainty in their data or analysis.
In summary, I think this study has some potential to contribute to the literature on the effect of surgical treatment of malignant tumors on walking speed and risk of falling, but it requires major revisions to address the methodological and analytical problems that I have identified. I suggest that the authors revise their manuscript accordingly and provide more details and clarity on their study design, methods, data, and analysis.
Author Response
Response to Reviewer 1:
We would like to take this opportunity to thank the Reviewer for their invaluable comments regarding our paper. We have made every effort to improve the manuscript. Below you will find detailed answers to all comments.
The study aimed to assess the effect of surgical treatment of the most common malignant tumors on walking speed and the risk of falling. I think this study has some merits and limitations, but it requires major revisions before it can be considered for publication.
Please find specific comments below.
- Introduction section needs a significant improvement.
The authors did not elucidate clearly why they chose to focus on four malignancies (breast, lung, colorectal and reproductive organ cancers) in their study. They should add their relative frequency, impact, or relevance to their research question. In addition, they should provide some statistics or evidence to support their claims. They could cite some sources that report the incidence, prevalence, mortality, and burden of cancer in different regions or countries. Is there no existing studies examining the relationship between surgical treatment, walking speed, and fall risk? The authors should provide more detail on previous studies that have addressed this topic, summarising the key findings and limitations of these studies and how this study addresses these issues.
We agree with the Reviewer and we made the change (rephrasing) in the Introduction section.
We added cancer epidemiology. We provided more detail on previous studies that have addressed this topic, we added a paragraph related to the effects of cancer and its treatment and we added a paragraph related to the findings of falls.
- Methods section is vague and lacks a lot of key information.
First, the authors used healthy people as a control group but not patients who did not undergo surgery as a control group. This makes it difficult to isolate the causal effect of surgery on the outcome. They should explain why they chose this design and how they ensured the comparability of the groups.
The authors opted for a group of healthy individuals because there was no possibility of ensuring a group of patients with a diagnosis of the specific cancer who had not undergone any treatment.
In the case of a malignant cancer diagnosis, it is essential to initiate treatment as quickly as possible, in the form of surgery or other forms of therapy (e.g. chemotherapy, radiotherapy, immunotherapy). At the same time, each of the chosen forms of treatment can impact the psychophysical condition. A review of the literature revealed changes in the walking and an increased risk of falls in patients undergoing systemic treatment.
Neoadjuvant treatment constituted a criterion of exclusion in the study.
Secondly,how did the authors randomise the participants for this study? Authors should provide a detailed description of the randomisation procedures and allocation hiding methods used in the manuscript. If the authors were completely randomly selected participants, what was done to ensure that they were sufficiently representative, this should also be elaborated in the manuscript.
We added information about recruitment and randomisation.
Thirdly, how did they standardise the gait assessment using the G-Walk accelerometer? The authors should provide more information on the validity and reliability of this device and software for measuring walking speed, as well as details on how they calibrated the device, instructed the participants, and controlled for any confounding factors that might affect gait performance during the study.
We added a paragraph related to BTS G Walk in the methods section. We have added information about device calibration, validity and reliability.
Fourthly, how did they administer the Falls Control Card to the participants? The authors should elaborate on how they collected and verified information about possible falls over a one-month period, as well as subjective feelings related to daily physical activity. They should also explain how they defined a fall and how they classified falls according to their causes, consequences and circumstances.
We added a paragraph related to the Falls Control Card in the methods section. We added a definition of fall and we classified falls according to their causes, consequences and circumstances.
Finally, authors should provide the clinical registration number and ethical review approval number in their methods section. I only found an Institutional Review Board Statement at the end of the manuscript and did not see any clinical registration number. If possible, I would suggest that they report any ethical issues or challenges they encountered during their study and how they resolved them. For example, how did they ensure the safety and well-being of their participants, especially those who underwent surgery for malignant tumors? Providing these information is essential for the quality and credibility of the paper and for the readers to evaluate the ethical conduct of the study.
The study received a positive opinion from the Senate Committee on Research Ethics at Wroclaw University of Health and Sport Sciences. Consent number: 28/2018. Approval date: 14/09/2018. The study was approved and was registered in the Lower Silesian Center of On-cology, Pulmonology, and Hematology in Wroclaw, Poland. Consent number: NDBI-106/18, NDBI 4/18/BN. This information is included in Ethics section (2.3).
All patients received medical and physiotherapeutic care.
- Results section has some gaps and errors that need to be corrected.
First, authors should add units to the data in tables, such as BMI in Table 1. This will make their data more clear and understandable for readers.
We added units to the data.
Secondly, footnotes need to be added to tables, such as what Study 1 represents. This will help readers follow logic and design of their study.
Footnotes have been added to all tables.
Thirdly, * in table already represents p<0.05, so why is it written in table as p<0.0001? This is redundant and confusing. They should either use * or exact p-value, but not both. They should also use consistent format for reporting p-values throughout paper, such as using two or three decimal places.
Throughout the work, the p-value is reported uniformly to four decimal places. A significance level of p<0.05 was accepted, and it was denoted with an asterisk (*) in the tables. The p-value reported in the tables as <0.0001 does not indicate the adopted level of significance but rather the actual value for a given analysis. It means that the calculated level of significance was lower than four decimal places (e.g. 0.0000002), and when rounded, this value would be 0 (0.0000), which is not accurate. Therefore, it was decided to present the data in this manner.
Fourthly, data in Table 3 and Table 4 are difficult to understand, with only p-values in one table. They should provide some descriptive statistics, such as means and standard deviations, for each group and each time point. They should also report effect sizes and confidence intervals for their tests of significance. This will allow readers to evaluate magnitude and precision of their results.
Tables 3 and 4 do not include measurement data (mean values and SD of walking speed) because they are a continuation of Table 2, which presents this data.
Table 2 shows the mean values of walking speed for each group in each study, along with standard deviations. Table 2 also contains the p-value of the ANOVA significance coefficient describing the significance of differences between groups and studies. Table 2 also presents effect size values for both analyses.
Table 2 demonstrates that the differences between groups and successive studies are significant. Therefore, post-hoc tests were conducted to examine in detail between which pairs of measurements (Table 4) and which groups the differences are significant (Table 3). The effect size values for these analyses are included in Table 2 in the Repetition column for Table 3 and the Group column for Table 4. Combining Tables 2, 3, and 4 is unfortunately not possible due to their size.
Fifthly, what is ch2 in Table 5 and ±95CI in Table 6? They should check writing throughout. They should use proper symbols and abbreviations for their statistical terms, such as χ2 and ± . They should also define any abbreviations that they use in their paper, such as OR (odds ratio) and CI (confidence interval).
Abbreviations are defined.
Sixthly, value of OR for walking speed in Table 6 is 0.002, while value of OR for walking speed in Table 7 is 333.96. First of all, these two results are a bit suspicious. In addition, p-value is surprisingly same for both. There is a relationship between p-value and OR-value. I cannot understand why an OR value of 0.002 would show such a small p-value. They should explain how they calculated OR and p-value for walking speed and check for any errors or outliers in their data or analysis. Finally, 95% confidence range for OR value of walking speed in Table 7 is 32.96-3844.27. It is clear that maximum and minimum values are very much separated from each other. Therefore, OR value may not be credible. They should report standard error or standard deviation of OR value and check for any sources of variability or uncertainty in their data or analysis.
The value OR=0.002 in Table 6 pertains to walking speed recorded in study 2, and the value OR=355.96 in Table 7 pertains to the difference in walking speed measured in studies 1 and 2. Both results are related, hence the p-value calculated for OR in both analyses is the same and it is exactly 0.000001 (adjusted to the reporting format <0.0001).
The Grubbs test was used to check the raw data for outliers. No outliers were found in studies 1 and 2. In study 3, there were 3 outliers out of 176 measurements, and in the difference b1-b2, there were 4 outliers out of 176 measurements.
Regression models were rechecked: the first one (Table 6) remained unchanged. The second one (Table 7) showed slightly different OR values for the difference in walking speed, but they did not change the results' interpretation. Changes were made to the text. The reporting of the logistic regression was also altered by adding the β coefficient, the β error, and the Wald statistics value.
In summary, I think this study has some potential to contribute to the literature on the effect of surgical treatment of malignant tumors on walking speed and risk of falling, but it requires major revisions to address the methodological and analytical problems that I have identified. I suggest that the authors revise their manuscript accordingly and provide more details and clarity on their study design, methods, data, and analysis.
Additionally, references were added and corrected.
We hope the revised version is now suitable for publication.
Thank you again for the review.
Yours faithfully, Authors
Reviewer 2 Report
Comments and Suggestions for Authors
Dear editor, thank you for the opportunity to review this study, which presents an interesting idea for medical and Physiotherapy professionals. My considerations about the manuscript are below:
Abstract:
- I believe that adding subtopics (Background, objectives, methods, results and conclusion) will help better understand the text. I suggest including these subtopics in the abstract.
- The introduction of the abstract is meaningless, with short sentences that are disconnected from subsequent sentences. I suggest that the authors rewrite the entire introduction of the abstract, making it more coherent with the problem of the study.
- The authors do not mention that they will investigate falls, however, they appear in the results. I suggest that the authors add how they evaluated the drops in the abstract methods.
- We cannot establish a relationship between cause and effect from a cross-sectional study. Therefore, the authors cannot conclude that the reduction in gait speed and the increase in falls was caused by the surgery. Authors should improve the conclusion of the study in the abstract.
Introduction:
- The introduction has very large paragraphs. I suggest segmenting them.
- The authors present in the introduction that scars or adhesions in soft tissue structures can reduce the walking speed of volunteers. However, they forget some myths and fears of patients, such as the fear of opening surgery stitches, the belief that post-surgery the best thing to do is to rest, these would be important points to expose, to better explain the problem.
- Regarding the objective of the study: I am unable to observe the impact of surgery on gait speed in a cross-sectional study. I can't make this relationship between cause and effect. I suggest that authors, both in the abstract and at the end of the introduction, organize the objective of the study.
Methods:
- What is the design/type of the study?
- How was the sample recruited? Was there a calculation to estimate the sample size, or is it a convenience sample? The authors need to further detail how the sample for this study was recruited.
- What is the objective of randomization? There is no way that each participant could have been randomized because, for example, there was a risk that a patient in the control group would end up in the breast cancer group, for example. What is the objective of this randomization? For me it doesn't make sense.
- Is this a cohort study? Authors need to make the type of study clear.
- Some information is important, such as whether patients practiced physical activity, whether they took any medication that lowered the level of consciousness, or for pain or muscle relaxants. Presence of vertigo and dizziness, muscle strength, all these aspects can influence the results if present in patients.
Results:
- I suggest that authors place in the footer of table 1 what each letter related to the groups means. This helps readers understand.
Discussion:
- The discussion is the poorest part of the study, in scientific terms.
- In the first four paragraphs of the discussion, the authors bring concepts of what walking is, what post-surgery is like and provide data. However, they do not provide a physiological explanation of how the surgical procedure itself can change gait speed, as they found, and this needs to be justified with scientific evidence.
- The same occurs in the discussion regarding the findings of falls. Falls are a multifactorial outcome. Therefore, it is difficult to say that a single factor caused the falls.
- The discussion also needs to justify how the surgical procedure itself can lead to falls, as they found and this needs to be justified with scientific evidence.
- The discussion itself needs to be greatly improved and based on evidence about these two outcomes: gait speed and falls.
- I also suggest that the authors write a paragraph mentioning how fears and myths related to surgery may have contributed to the results found. Since the results show that the lower in the body, that is, the closer the surgery was to the lower limbs, the gait speed was reduced. Therefore, there may be a relationship between fear and proximity to the surgical incision and gait speed and falls.
- I also suggest, adding to the limitations, that as no calculation was made to estimate the sample size of the study, these results cannot be generalized.
Conclusions:
- I believe that the conclusions are very incisive and hasty, especially because the authors did not bring into the discussion an evidence model that explains how the procedure itself can lead to reduced gait and cause falls in study volunteers.
- I suggest that the authors better organize the conclusion of the study.
Author Response
Response to Reviewer 2:
We would like to take this opportunity to thank the Reviewer for their invaluable comments regarding our paper. We have made every effort to improve the manuscript. Below you will find detailed answers to all comments.
Abstract:
- I believe that adding subtopics (Background, objectives, methods, results and conclusion) will help better understand the text. I suggest including these subtopics in the abstract.
We included subtopics.
- The introduction of the abstract is meaningless, with short sentences that are disconnected from subsequent sentences. I suggest that the authors rewrite the entire introduction of the abstract, making it more coherent with the problem of the study.
We agree with the Reviewer and we made the change (rephrasing) in the Introduction section of the abstract.
- The authors do not mention that they will investigate falls, however, they appear in the results. I suggest that the authors add how they evaluated the drops in the abstract methods.
We have completed the description.
- We cannot establish a relationship between cause and effect from a cross-sectional study. Therefore, the authors cannot conclude that the reduction in gait speed and the increase in falls was caused by the surgery. Authors should improve the conclusion of the study in the abstract.
We improved the conclusion of the study in the abstract.
Introduction:
- The introduction has very large paragraphs. I suggest segmenting them.
we introduced a division.
- The authors present in the introduction that scars or adhesions in soft tissue structures can reduce the walking speed of volunteers. However, they forget some myths and fears of patients, such as the fear of opening surgery stitches, the belief that post-surgery the best thing to do is to rest, these would be important points to expose, to better explain the problem.
We agree with the Reviewer and we made the change (rephrasing) in the Introduction section, we added a paragraph related to pain and anxiety and added new references.
- Regarding the objective of the study: I am unable to observe the impact of surgery on gait speed in a cross-sectional study. I can't make this relationship between cause and effect. I suggest that authors, both in the abstract and at the end of the introduction, organize the objective of the study.
We clarified the objective.
Methods:
- What is the design/type of the study?
We have added the design/type of the study.
- How was the sample recruited? Was there a calculation to estimate the sample size, or is it a convenience sample? The authors need to further detail how the sample for this study was recruited.
Sample size calculations were not performed. To confirm whether the number of subjects was sufficient, the power of the ANOVA test for the walking speed in the study groups was checked. The obtained test power was close to 1, confirming the adequate sample size. We added the information in the table 2.
- What is the objective of randomization? There is no way that each participant could have been randomized because, for example, there was a risk that a patient in the control group would end up in the breast cancer group, for example. What is the objective of this randomization? For me it doesn't make sense.
The authors have improved the translation, we hope that the revised description of group recruitment is now clear.
- Is this a cohort study? Authors need to make the type of study clear.
We have added the design/type of the study.
- Some information is important, such as whether patients practiced physical activity, whether they took any medication that lowered the level of consciousness, or for pain or muscle relaxants. Presence of vertigo and dizziness, muscle strength, all these aspects can influence the results if present in patients.
We have made the exclusion criteria more specific to include medications.
We added the following information: Throughout the study, patients were required to maintain their usual lifestyle.
Results:
- I suggest that authors place in the footer of table 1 what each letter related to the groups means. This helps readers understand.
We have added the legend in the footer to the tables.
Discussion:
- The discussion is the poorest part of the study, in scientific terms.
We agree with the Reviewer and completed the discussion, and added new references.
- In the first four paragraphs of the discussion, the authors bring concepts of what walking is, what post-surgery is like and provide data. However, they do not provide a physiological explanation of how the surgical procedure itself can change gait speed, as they found, and this needs to be justified with scientific evidence.
We added a paragraph related to the effects of cancer and its treatment in the introduction section.
We also added a paragraph on the impact of surgical interventions on walking speed in the discussion section.
- The same occurs in the discussion regarding the findings of falls. Falls are a multifactorial outcome. Therefore, it is difficult to say that a single factor caused the falls.
- The discussion also needs to justify how the surgical procedure itself can lead to falls, as they found and this needs to be justified with scientific evidence.
We added a paragraph related to the findings of falls in the introduction.
we also added a paragraph on the impact of cancer treatment on falls in the discussion section.
- The discussion itself needs to be greatly improved and based on evidence about these two outcomes: gait speed and falls.
We agree with the Reviewer and we made the change (rephrasing) in the Discussion section.
- I also suggest that the authors write a paragraph mentioning how fears and myths related to surgery may have contributed to the results found. Since the results show that the lower in the body, that is, the closer the surgery was to the lower limbs, the gait speed was reduced. Therefore, there may be a relationship between fear and proximity to the surgical incision and gait speed and falls.
We added a paragraph related to pain and anxiety in the Introduction section.
- I also suggest, adding to the limitations, that as no calculation was made to estimate the sample size of the study, these results cannot be generalized.
To confirm whether the number of subjects was sufficient, the power of the ANOVA test for the walking speed in the study groups was checked. The obtained test power was close to 1, confirming the adequate sample size.
Conclusions:
- I believe that the conclusions are very incisive and hasty, especially because the authors did not bring into the discussion an evidence model that explains how the procedure itself can lead to reduced gait and cause falls in study volunteers.
- I suggest that the authors better organize the conclusion of the study.
We made the change (rephrasing) in the Conclusion section.
Additionally, references were added and corrected.
We hope the revised version is now suitable for publication.
Thank you again for the review.
Yours faithfully, Authors
Reviewer 3 Report
Comments and Suggestions for Authors
Authors perfomed an observational study in order to evaluate the risk of falling and the gait performance of cancer patients after surgery. This topic is of interest, as gait is fundamental for an optimal quality of life and the interaction with others.
Introduction: Please focus on gait and detail the consequences of the surgery. Does radiotherapy and chemotherapy may also affect this outcome? Moreover, I think there are some references missing on this section.
Methods: How was the sample size calculated for this study? It is not indicated if the patients signed an informed consent before their participation on the study. Please clarify this information.
Results: please indicate a footnote after each tables in order to explain all abreviations (BU, G, DS, etc). It would be interesting to include the mean and SD of the whole sample.
These results are well discussed, but finally in the conclusion, I would rather write a single paragraph, explaining both conclusions answering the objectives of this study
Author Response
Response to Reviewer 3:
We would like to take this opportunity to thank the Reviewer for their invaluable comments regarding our paper. We have made every effort to improve the manuscript. Below you will find detailed answers to all comments.
Authors perfomed an observational study in order to evaluate the risk of falling and the gait performance of cancer patients after surgery. This topic is of interest, as gait is fundamental for an optimal quality of life and the interaction with others.
Introduction: Please focus on gait and detail the consequences of the surgery. Does radiotherapy and chemotherapy may also affect this outcome? Moreover, I think there are some references missing on this section.
We agree with the Reviewer and we made the change (rephrasing) in the Introduction section. We added a paragraph related to the effects of cancer and its treatment on walking speed and falls. We added new references.
Methods: How was the sample size calculated for this study? It is not indicated if the patients signed an informed consent before their participation on the study. Please clarify this information.
Sample size calculations were not performed. To confirm whether the number of subjects was sufficient, the power of the ANOVA test for the walking speed in the study groups was checked. The obtained test power was close to 1, confirming the adequate sample size. We added the information in the table 2.
Informed consent of the patient to participate in the study is included in the study inclusion criteria, point 5.
Results: please indicate a footnote after each tables in order to explain all abreviations (BU, G, DS, etc). It would be interesting to include the mean and SD of the whole sample.
We have added the legend in the footer to the tables.
We included the mean and SD of the whole sample (table 2).
These results are well discussed, but finally in the conclusion, I would rather write a single paragraph, explaining both conclusions answering the objectives of this study
We made the change (rephrasing) in the Conclusion section.
Additionally, references were added and corrected.
We hope the revised version is now suitable for publication.
Thank you again for the review.
Yours faithfully, Authors
Round 2
Reviewer 2 Report
Comments and Suggestions for Authors
Congratulations to the authors, the changes in the manuscript made the text clearer for readers to understand.
Author Response
Response to Reviewer 2:
We would like to take this opportunity to thank the Reviewer for their review our paper.
Yours faithfully, Authors
Reviewer 3 Report
Comments and Suggestions for Authors
Authors have improved the quality of their manuscript by following the reviewers' recommendations.
In my opinion, now I would only recommend the authors to use the STROBE checklist for this cohort and include it as an appendix. And please revise all your grammar, as "chi squer" is written in page 6, line 229
Author Response
Response to Reviewer 3:
We would like to take this opportunity to thank the Reviewer for their review our paper.
Authors have improved the quality of their manuscript by following the reviewers' recommendations.
In my opinion, now I would only recommend the authors to use the STROBE checklist for this cohort and include it as an appendix. And please revise all your grammar, as "chi squer" is written in page 6, line 229
We added STROBE checklist for this cohort and we included it as an appendix.
According to the STROBE checklist, we added in the manuscript:
- study’s design in the abstract
- periods of recruitment in the method section
We corrected "chi squer test" to "Chi-square test"
Thank you again for the review.
Yours faithfully, Authors